# Surgical Excision of an Extratesticular Anaplastic Carcinoma in a Variable Kingsnake (*Lampropeltis mexicana*)

**DOI:** 10.3390/ani14060874

**Published:** 2024-03-13

**Authors:** Viola Zappone, Claudia Rifici, Matteo Marino, Manuel Morici, Giada Giambrone, Filippo Spadola

**Affiliations:** 1Department of Veterinary Sciences, University of Messina, Viale Palatucci, 13, 98168 Messina, Italy; viola.zappone@unime.it (V.Z.); claudia.rifici@unime.it (C.R.); giada.giambrone@studenti.unime.it (G.G.); fspadola@unime.it (F.S.); 2Pombia Safari Park, 28050 Pombia, Italy

**Keywords:** *Lampropeltis mexicana*, surgical excision, extratesticular anaplastic carcinoma, snake

## Abstract

**Simple Summary:**

An adult male variable kingsnake was examined for a three-week history of anorexia and body deformities, revealing poor condition and an intracoelomic mass. With the owner’s consent, an exploratory celiotomy was performed to remove the mass, which was identified as an undifferentiated tumour by modified Wright–Giemsa stain smears. Histological examination revealed a solid proliferation of highly tubular anaplastic cells and plurinucleated cells positive for cytokeratin and vascular endothelial growth factor, but not vimentin. Periodic acid–Schiff staining showed large granular cells characteristic of efferent ducts. A diagnosis of extratesticular anaplastic carcinoma was made as the first reported case in the male genital apparatus of snakes.

**Abstract:**

An adult male variable kingsnake (*Lampropeltis mexicana*) was presented for examination due to a three-week history of anorexia and obvious body deformities. On objective examination the animal was in poor condition, and on palpation, an intracoelomic mass was noted approximately in the distal third of the body, cranial to the cloaca. In agreement with the owner, an exploratory celiotomy was planned and performed and the mass was surgically removed. Modified Wright–Giemsa stain impression smears were taken, which were consistent with an undifferentiated tumour. Histological examination revealed the presence of a solid proliferation composed of highly tubular anaplastic cells and abundant multinucleated cells. The neoplastic cells were positive for cytokeratin (AE1/AE3), but not for vimentin. Periodic acid–Schiff (PAS) staining revealed the presence of large granular cells, which can be identified as the characteristic cells of the efferent ducts. Based on the morphological and immunohistochemical findings, the diagnosis of extratesticular anaplastic carcinoma was made. To the authors’ knowledge, this type of neoplasm has never been reported in the male genital apparatus of snakes.

## 1. Introduction

Based on animals sent to diagnostic pathology services, the incidence of neoplasia in reptiles ranges from 12% to 26% [1,2,3,4]. Variation in these estimates may be due to reporting bias based on differences in location, type of collection, and private studies conducting verification. Turtles and tortoises have a relatively low incidence of neoplasia (10.3%) [1,2,5]. The skin is the most commonly affected site (21.2%) [6,7]. Cutaneous squamous cell carcinoma is the most commonly observed neoplasm in this species, although mastocytomas, fibromas, and adenocarcinomas have also been described [5,6,7,8]. Saurians have a moderate incidence of neoplastic disease compared to snakes, turtles, and tortoises, with the haematopoietic, cutaneous, and hepatic systems most commonly affected [1,2,3,5]. Pogona vitticeps and iguanas are overrepresented in recent case studies, probably due to their popularity as pets and increased frequency of veterinary visits.

Snakes are the class of reptiles in which neoplasms are most often found [1,2,3,5,9]. The haematopoietic and lymphoid systems are most affected, followed by neoplasms of epithelial origin [9,10], of which renal adenomas and adenocarcinomas appear to have a high incidence. Among skin tumours, melanophoroma is the third most common type of tumour reported in snakes, usually presenting as a single skin mass [4,11]. Among the adenomas and renal adenocarcinomas examined in eight snakes, most tumours were characterised by a low mitotic index and a locally invasive nature [12]. Melanophoromas are the third most common type of tumour reported in snakes, usually presenting as a single skin mass. These tumours are locally invasive and have a poor prognosis [4,11,13]. Ovarian adenocarcinomas have been described in several snake species and are typically associated with diffuse metastases and a poor prognosis. These tumours may have a hormonal predisposition, although the cause has not been established [14]. In snakes, neoplasms affecting the reproductive system are most common in the females, with a high incidence of granulosa cell tumours [15]. Macroscopically, these neoplasms have a smooth, regular surface with a cystic appearance on sectioning [16]. Numerous other tumour types have been found in the female genital tract of snakes, including stromal tumours of the sex cord, carcinomas, and mesenchymal neoplasms of the ovary, including haemangiomas, fibromas, and sarcomas [9,17]. Neoplasms of the male genital tract have a lower incidence than those of the female reproductive tract. Tumour types include Leydig cell tumours, seminomas, Sertoli cell tumours, and sarcomas [16,18]. Interstitial cell tumours, also known as Leydig cell tumours, have been reported in snakes, saurians, and chelonians. Histologically, there is expansion of the testicular interstitium by polygonal or occasionally elongated cells with eosinophilic cytoplasm, which have a “ground glass” appearance typical of interstitial or “Leydig” cells. This type of tumour has been found in a Burmese python, *Epicrates* sp., and a Vipera palaestinae [9,15,17]. Seminomas are neoplasms of the germ cells of the testis and are uncommon in reptiles. Microscopically, they consist of intratubular or diffuse sheets of neoplastic round cells with sparse stroma. The morphology of the neoplastic cells is typical of germ cells—large, round cells with a large round vesicular nucleus, a large nucleolus, and a small amount of eosinophilic cytoplasm. Multinucleated cells and small lymphocytes are often scattered throughout the neoplasm [17]. This type of tumour has been found in an Indian cobra [17]. Sertoli cell tumours are also rare neoplasms in reptiles. Macroscopically, these neoplasms appear firm due to the large amount of scirrhous stroma. Sertoli cell tumours are distinguished from other primary testicular neoplasms by the histomorphology of their cells and the abundance of scirrhous stroma. In well-differentiated variants, neoplastic cells resemble mature Sertoli cells, palisading along the stroma, forming small tubules with occasional cystic dilatation, and are elongated with foamy, pale eosinophilic cytoplasm and small basal nuclei. In less differentiated Sertoli cell tumours, the nuclei are pleomorphic and the cells are less likely to palisade. There are very few data on Sertoli cell tumours in reptiles, and all reports in the literature are in snakes. This neoplasm has been reported in a Pantherophis guttatus, a Lampropeltis triangulum hondurensis, a Thamnophis elegans, and a Naja haje [2,17].

## 2. Case Description

An 8-year-old adult male variable kingsnake (*Lampropeltys mexicana*) was brought to the Veterinary Teaching Hospital of the University of Messina (Italy) for examination due to the presence of a deformity in the distal third of the body cranially at the cloacal opening. The subject was kept in a terrarium, fed live mice weekly, moulted regularly, and successfully bred in the year before presentation. The snake had a three-week history of anorexia, failure to defecate for several weeks, and depression. The subject then underwent ultrasound, which revealed a moderately echogenic intracelomatic mass. In agreement with the owner, an exploratory celiotomy was planned and performed, and the mass was surgically removed.

A total of 1.25 mL of blood was collected from the caudal vein into a test tube containing sodium citrate. Before surgery, a blood smear stained with Fast Quick—M.G.G. was made, which allowed us to evaluate for packed cell volume, while uric acid, alkaline phosphatase, glucose, calcium, phosphorus, sodium, and potassium were obtained by biochemical analysis (mnchip celercare v5). All clinical chemistry parameters were within the normal ranges reported for *Pantherophis guttatus* (Table 1) [19]. Due to the absence of bibliographical references on the haematology of *Lampropeltis mexicana*, we decided to use the values published on *Pantherophis guttaus* as the two species are similar to each other; in fact, they share the same natural habitat, have the same captive housing, and, by mating, are capable of producing fertile offspring.

The air temperature in the anaesthetic room was set at 26 °C. Basal heart rate was recorded using a vascular Doppler probe (PD1v Pocket Vascular Doppler, Ultrasound Technologies, Caldicot, UK), and basal respiratory rate was assessed by monitoring the body wall expansions of the snake at rest. The subject was premedicated with alfaxalone (Alfaxan, 10 mg/mL, Vetoquinol, Magny-Vernois, France) [20], which was administered intravenously via the ventral tail vein [21,22] at a dose of 10 mg/kg using a 2.5 mL syringe with a 26 G needle (PIC, Ponte San Giovanni (Perugia), Italy). Alfaxalone was administered as a single bolus over a few seconds. The subject was maintained with isoflurane (Isoflo^®^250 mL, Zooetis Italia, Roma, Italy) after tracheal intubation.

After careful preparation of the surgical field with diluted antiseptic solution (Betadine^®^ 7.5%, l h iodo 7.5, Lombarda h. Srl, Abbiategrasso, Italy), a celiotomy was performed near the external deformity. Blunt dissection was performed laterally to the ventral midline, where the coelomic membrane was incised to reach the coelomic cavity. Within the coelomic cavity, an approximately 5 cm long, vascularized, adipose-tissue-covered neoformation was visualized near the kidney. The blood vessels were ligated, and the mass was completely excised. The coelomic membrane was carefully sutured with 4-0 polyglycate (Vicryl (polyglactin 910) by Ehticon) in a simple continuous pattern. The skin was sutured with 4-0 polyglycate (Vicryl (polyglactin 910) by Ehticon) in an everting horizontal mattress pattern because reptilian skin has a strong tendency to invert and create incision edges that are not in apposition [23].

After surgical resection, which resolved the bowel obstruction, postoperative therapy included both systemic antibiotic therapy with enrofloxacin (Baytril^®,^ Elanco Italia S.p.A., Sesto Fiorentino, Italy) 10 mg/kg IM every 48 h for 1 week and analgesia with 5 mg/kg tramadol (Tramal^®^ 50, 50 mg/mL, Grunenthal GMbH, Aachen, Germany) IM every 24 h for 5 days.

The animal was kept under observation for one day after surgery, after which it was returned to its owner. A telephone follow-up was carried out at 30 and 60 days post-surgery. In both follow-ups, the owner reported that the snake was in excellent health, feeding spontaneously, and performing its usual activities.

The excised tissues were sent to the histopathology laboratory of the Department of Veterinary Sciences, University of Messina, Messina, Italy, where, after macroscopic and cytological evaluation by staining with MGG-Q, they were fixed in 10% buffered formalin and routinely embedded in paraffin. Histological sections 3–4 μm thick were stained with hematoxilin–eosin. Based on the histological images observed, it was necessary to perform histochemical and immunohistochemical investigations to characterise tumour site and histotype.

The sections were subjected to histochemical staining with periodic acid–Shiff (PAS) and immunohistochemistry with vimentin, cytokeratin, and chromogranin. For immunohistochemistry, 5 μm thick tissue sections were mounted on polylaminated slides and incubated at 37 °C. The method included antigen unmasking by microwave treatment in 0.01 mol/L citrate buffer, pH6, for 15 min. Endogenous peroxidase activity was blocked with 0.3% hydrogen peroxide in methanol for 30 min, while nonspecific protein reactions were inhibited after incubation with 2.5% BSA (bovine serum albumin) for 30 min.

The slides were then incubated overnight at 4 °C with the following primary antibodies: anti-pancytokeratinAE1–AE3 (mouse monoclonal; 1:200 dilution, Santa Cruz Biotechnology, Dallas, TX, USA), anti-Vimentin (Vimentin (monoclonal antibody 1/200, Santa Cruz, Dallas, TX, USA)), and chromogranin (mouse monoclonal; 1:200 dilution, Bio Optica, Milano, Italy).

They were then incubated for 30 min at room temperature with biotinylated secondary antibodies specific for the host species and isotype of the primary antibody (goat anti-rabbit biotinylated IgG, dilution 1: 200, Biospa, Milano, Italy; goat anti-mouse biotinylated IgG, dilution 1:200, Biospa, Milano, Italy). The reaction was then developed with acetylated avidin (dilution 1:250, Biospa, Milano, Italy) and detected with DAB (3,3′-diaminobenzidine, Dako Laboratories, Santa Clara, CA, USA) and counterstained with Carazzi hematoxylin.

Negative controls were also performed by omission of primary Abs, the substitution of primary Abs with normal IgGs, and substitution of primary Abs with nonreactive antibodies of the same species and immunoglobulin class. The immunohistochemical stain was interpreted by assessing the intensity of labelling. Cytoplasmic and/or membrane immunoreactivity was considered positive.

## 3. Results

Upon macroscopic examination, the mass had a globular shape with irregular margins, measuring approximately 6 × 5 × 5 cm, covered by an adherent fibrous membrane. On section, it had a multinodular and variegated appearance ranging from light brown to dark red, with multiple necrotic foci and a fibrous to gelatinous consistency (Figure 1).

Upon cytological examination, moderate cellularity was observed, consisting mainly of small but visible cell clusters, sometimes with an exfoliative pattern of a pseudotubular or acinar nature, composed of neoplastic cells aligned along processes of mature collagenous connective tissue. The cells, sometimes markedly anaplastic, were fusiform to oval, with a nucleus containing punctate to small clumps of chromatin, often with prominent and multiple nucleoli, and intensely basophilic or finely vacuolated cytoplasm. There was marked anisocaryosis and anisocytosis and the presence of multinucleated giant cells and vacuolated macrophages. Large cells with granular cytoplasm were present in the context.

Histological sections stained with EE showed a highly cellular neoplasm consisting of a poorly differentiated population of predominantly spindle to polygonal cells with abundant bi- and multinucleated cells. The cells were medium to large in size. Anisocytosis and anisocaryosis were noted. Irregular, often multiple, prominent nucleoli were observed. Moreover, thanks to the periodic acid–Schiff (PAS) stain, it was possible to observe the presence of large granular cells characteristic of efferent ductules (Figure 2). Based on the cytological and histological appearance, a poorly differentiated extratesticular neoplasm was diagnosed.

Immunohistochemistry revealed the epithelial origin of the neoplasm with a positive and specific reaction to cytokeratins (Figure 3). The absence of vimentin also excluded the presence of areas of mesenchymal differentiation (mesenchymal shift).

Based on histological and immunohistochemical studies, an anaplastic carcinoma of the extratesticular region (sexual segment of the kidney) was diagnosed in a *Lampropeltis mexicana* snake.

## 4. Discussion

This report describes the clinical and pathological findings of a tumour in a male reproductive system in a snake. The male reproductive apparatus in snakes is very complex. Snakes and other squamates have a system of efferent ducts/tubules that have a mixed function, both for the passage of spermatozoa and glandular-like with a secretory function (accessory sex glands) and for the “storage” of sperm. These are androgen-dependent accessory sex glands that secrete the products necessary for viable sperm [24]. The seminiferous tubules empty into the vas deferens, which have an epithelial structure with both simple cuboidal and ciliated cuboidal cells. The nonciliated cubic cells have secretory capacity and are filled with many PAS-positive secretory granules that discharge directly into the ureter at different times of the year [25]. The size of the secretory granules of the renal epithelium peaks at the time of mating and regresses in association with post-spermatogenesis regression [24]. The efferent ductulae are, therefore, an extratesticular portion in continuity with the testicular network and continue with the epididymal duct, which drains into the ureter [26]. Based on these anatomical considerations, the location of this testicular portion and the presence of PAS-positive granulations in the cytoplasm of some large cells, it was possible to identify the origin of the tumour as the intermediate portion between the testis and the kidney (“epithelial portion”), i.e., the area that, in reptiles, starts from the caudal portion of the kidney and continues with the testis.

The cytopathological features suggested only a marked malignancy of the neoformation and, in addition, it was extremely difficult to define the neoplastic origin macroscopically because the voluminous tumour mass altered the topography of the coelom.

Upon observation of the cell shape and cell arrangement at low-power magnification, we initially suspected an undifferentiated tumour. Nevertheless, the anaplastic cell features required further investigation by immunohistochemistry to define the exact cell origin. Immunohistochemical analysis of reptilian tumours has not been routinely described in the literature due to few commercially available antibodies for reptilian species, but the antibodies we used can be considered valid because they have been reported in the literature for other types of tumours in this species. The pancytokeratin we used (CK AE1–AE3) has been widely used to define tumours in reptiles. In contrast, a positive immunohistochemical reaction in neoplastic cells using pancytokeratin has been observed in cases of ovarian cystadenocarcinoma and metastatic heart cancer cells in an *Iguana iguana* and the superficial epithelium of an iguana ovary, in ovarian tissue of the lizard *Podarcis sicula*, and recently, Alibardi and Toni (2005) [27] also showed that snake epidermal layers use proteins with epitopes in common with those of the mammalian epidermis via pancytokeratin [28].

The antibodies vimentin and chromogranin have been widely used in neuroendocrine tumours in snakes [29], so their negative expression in our case allows us to exclude a neuroendocrine neoplasm or APUD-derived or GIST-like mesenchymal.

In this paper, we described a case of anaplastic carcinoma of the extratesticular region in a snake, *Lampropeltis mexicana*, representing a unique occasional finding, as reports of tumours of epithelial origin of the male spermatic duct or renal accessory sex segments are absent in the literature to date [29]. Moreover, the search for new and specific immunohistochemical markers, allowing more precise diagnostic and prognostic indications, should be stimulated by the complex anatomy and physiology of these species and the difficulty in distinguishing between the different tumour types. The increase in the number of neoplasms described in reptiles in recent years highlights the growing interest in this class of animals as companion animals. It also underlines the importance of collaboration between the clinician and the pathologist, a synergy that is essential for a better diagnostic and therapeutic approach.

To the authors’ knowledge, this is the first case of neoplasia of the sexual accessory segments of the male reproductive system described in a snake.

## 5. Conclusions

In conclusion, this case report highlights the presentation, diagnosis, and successful surgical management of an extratesticular anaplastic carcinoma in an adult male variable kingsnake (*Lampropeltis mexicana*). This report contributes to the limited literature on neoplasms of the male genital apparatus in snakes, with extratesticular anaplastic carcinoma being a novel finding in this species. Further research and documentation are warranted to improve our understanding of such rare pathological entities in reptile medicine.

## Figures and Tables

**Figure 1 animals-14-00874-f001:**
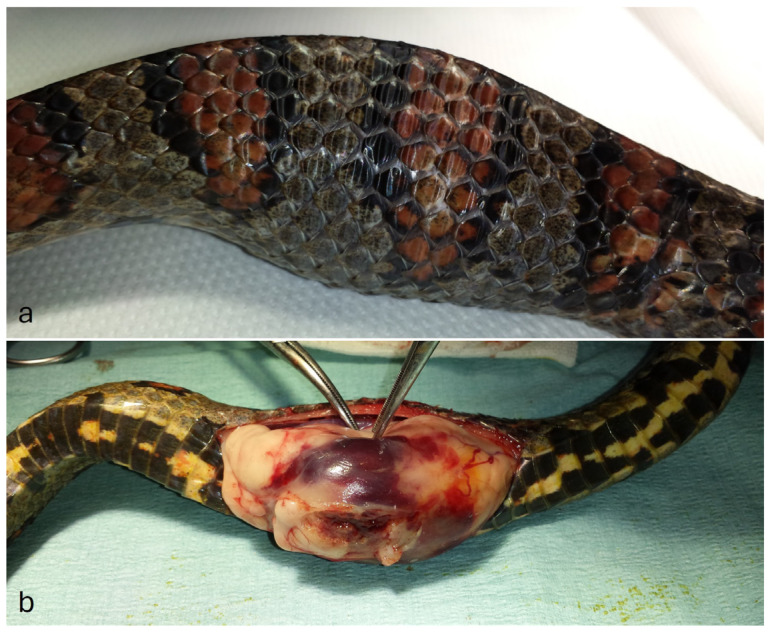
(**a**) Deformity in the distal third of the body cranially at the cloacal opening. (**b**) Macroscopic appearance of the intracoelomic mass at the time of surgical excision.

**Figure 2 animals-14-00874-f002:**
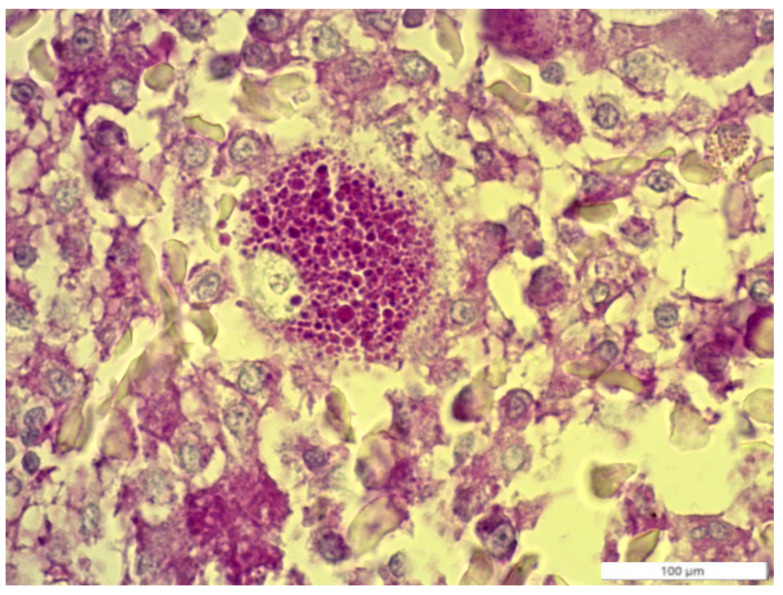
Positive granular cells of efferent ductules. PAS; 100 µm.

**Figure 3 animals-14-00874-f003:**
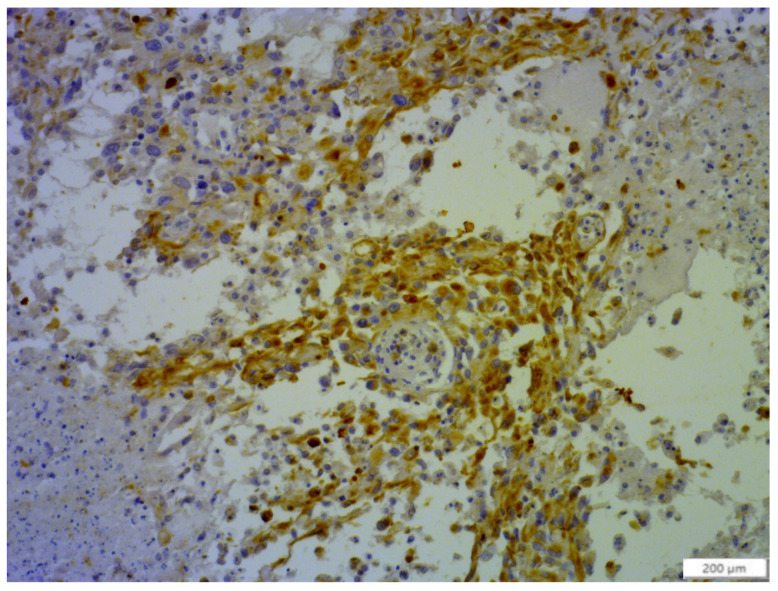
Cytokeratin positivity in neoplastic fusiform cells; 200 µm.

**Table 1 animals-14-00874-t001:** Comparison between the parameters we obtained and those followed as a guideline.

Parameter	Zims, 2019 [19]	Our Parameter
Packed cell volume (/1)	0.16–0.44 (mean 0.31)	0.29
Glucose (mmol/L)	0.6–5.1 (mean 2.9)	3.08
Calcium (total) (mmol/L)	2.4–5.3 (mean 4.0)	7.00
Phosphorus (mmol/L)	0.58–2.31 (mean 1.26)	1.76
Sodium (mmol/L)	148–178 (mean 162)	>170
Potassium (mmol/L)	1.5–9.8 (mean 5.1)	6.68
Uric acid (umol/L)	16–894 (mean 342)	294
AST (IU/L)	0–117 (mean 28)	32

## Data Availability

Data are contained within the article.

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
