# Peer review of "Surgical Excision of an Extratesticular Anaplastic Carcinoma in a Variable Kingsnake (Lampropeltis mexicana)"

_animals, 2024, doi:10.3390/ani14060874_

Round 1
Reviewer 1 Report
Comments and Suggestions for Authors
This manuscript describes a case report of a novel neoplasm in a single male variable kingsnake. This is a very interesting case and although it is a single report, the novelty of the neoplasm does allow for the opportunity for publication. There are a few minor edits that should be corrected and one major concern that needs to be addressed before this report should be considered for publication.
The major concern in this case report is the apparent inadequate aseptic technique intra-operatively. Figure 1b appears to be an intraoperative photo. The animal is not draped for celiotomy, which does not adhere to proper aseptic surgical technique. Also, based on the patient positioning, it appears as if the patient was moving during the procedure. It is unusual for a fully anesthetized snake to have an acute flexion as seen 6-8 ventral scales to the left of the incision. Hopefully, this is just odd positioning by the surgeon. It would be good to hear an explanation for these deficits.
Some additional concerns:
Lines 3, 10, 51, 202: "Variable" should not be capitalized.
Lines 3, 31, 48, 196: "Mexicana" should not be capitalized.
Lines 13, 22: "MGC-Quik" should be listed as a "modified Wright-Giemsa stain"
Lines 39-40: "the female sex" would be better stated as "females"
Line 54: using the term "theca" is confusing. Please rephrase.
Line 56: "culling" is typically used as a synonym for euthanasia. Please rephrase.
Lines 56-57: Providing an image of the ultrasound would really enhance this case, since it is a novel neoplasm.
Lines 60-62: Please provide the actual values for these parameters. There are no reference ranges for "snakes". Each species has their own reference range and most do not have published values. Additionally, normal ranges will vary depending on sex and season.
Lines 70, 71, 82, 87, 99-102, 104-107: Please provide brand, manufacturer, city, and state/country
Lines 82-84: Please mention whether these injections were given in the cranial or caudal half of the body. Most injections in reptiles are given in the cranial half but tramadol has been shown to be more efficacious if given in the caudal half.
Line 83: "analgesia" is a better term than "pain relief"
Lines 88, 92: "um" (micrometers) is preferable to "u" (microns)
Lines 94, 95, 97, 103: "min" is preferable to " ' "
Line 112: since this mass is a three-dimensional object, please provide the third measurement (length, width, height)
Line 118: "intracelomic" and not "intraceloma"
Lines 137, 145: please add scale bars to images
Line 158: delete "0"
Line 180: please mention the type of iguana and include genus and species
Line 181: please italicize "Podarcis sicula"
Comments on the Quality of English LanguageOverall, the English is good. A few minor edits were already mentioned. There are a couple of incidents where the spelling switches from American English to British English.
Author Response
Dear reviewer
Thank you for taking the time to review our manuscript.
Below you will find the point-by-point responses to your corrections / suggestions.
The major concern in this case report is the apparent inadequate aseptic technique intra-operatively. Figure 1b appears to be an intraoperative photo. The animal is not draped for celiotomy, which does not adhere to proper aseptic surgical technique. Also, based on the patient positioning, it appears as if the patient was moving during the procedure. It is unusual for a fully anesthetized snake to have an acute flexion as seen 6-8 ventral scales to the left of the incision. Hopefully, this is just odd positioning by the surgeon. It would be good to hear an explanation for these deficits.
The intraoperative image shows the animal in an abnormal position due to the displacement required to allow removal of the large mass. If the image needs to be edited for editorial reasons, it can be cropped.
Lines 3, 10, 51, 202: "Variable" should not be capitalized.
The authors followed the suggestion.
Lines 3, 31, 48, 196: "Mexicana" should not be capitalized.
The authors followed the suggestion.
Lines 13, 22: "MGC-Quik" should be listed as a "modified Wright-Giemsa stain"
The authors followed the suggestion.
Lines 39-40: "the female sex" would be better stated as "females"
The authors followed the suggestion.
Line 54: using the term "theca" is confusing. Please rephrase.
The authors followed the suggestion.
Line 56: "culling" is typically used as a synonym for euthanasia. Please rephrase.
The authors followed the suggestion.
Lines 56-57: Providing an image of the ultrasound would really enhance this case, since it is a novel neoplasm.
Unfortunately, the ultrasound image was lost due to a fault in the ultrasound machine.
Lines 60-62: Please provide the actual values for these parameters. There are no reference ranges for "snakes". Each species has their own reference range and most do not have published values. Additionally, normal ranges will vary depending on sex and season.
The authors followed the suggestion.
Lines 70, 71, 82, 87, 99-102, 104-107: Please provide brand, manufacturer, city, and state/country
The authors followed the suggestion.
Lines 82-84: Please mention whether these injections were given in the cranial or caudal half of the body. Most injections in reptiles are given in the cranial half but tramadol has been shown to be more efficacious if given in the caudal half.
The injection was performed on the cranial half of the body, as the authors had found this to be more effective than the caudal.
Line 83: "analgesia" is a better term than "pain relief"
The authors followed the suggestion.
Lines 88, 92: "um" (micrometers) is preferable to "u" (microns)
The authors followed the suggestion.
Lines 94, 95, 97, 103: "min" is preferable to " ' "
The authors followed the suggestion.
Line 112: since this mass is a three-dimensional object, please provide the third measurement (length, width, height)
The authors followed the suggestion.
Line 118: "intracelomic" and not "intraceloma"
The authors followed the suggestion.
Lines 137, 145: please add scale bars to images
The authors followed the suggestion.
Line 158: delete "0"
The authors followed the suggestion.
Line 180: please mention the type of iguana and include genus and species
The authors followed the suggestion.
Line 181: please italicize "Podarcis sicula"
The authors followed the suggestion.

Reviewer 2 Report
Comments and Suggestions for Authors
I found the manuscript interesting from a scientific and clinical point of view. However, some aspects need improvement:
Lines 61-62: include information about packed cell volume
Lines 90-92: indicate why the pathologist had to perform immunohistochemistry. include full name before PAS acronym. Revise this aspect throughout the manuscript for other acronyms.
Immunohistochemistry: Explain the type of validation you made for c-kit and VEGF (if any). Include the positive and negative controls you used for all the markers used. See additional comments in the discussion for c-kit and VEGF.
Fig 2 (define PAS), ampliation, the value is the total or objective??? include arrows. Figure 3: include arrows and full name, not acronyms. What is IIC? 20x????
Results: what happened to the snake? Is he alive? Did he fully recover? The authors spend a lot of time on material and methods describing surgery and did not include any data about recovery and follow-up in the results.
Discussion:
Lines 186-188 (about c-kit): This statement is a big limitation. I don't see the need to include the C-kit data. if so, you need to state the limitations and don't make any conclusion. What about if c-kit was positive? Would not be a carcinoma anymore? several carcinomas revealed c-kit positivity...
Lines 190-194: VEGF is not needed for diagnosis. In this context, VEGF immunolabelling doesn't bring any additional information, because you did not perform a new vessel immunolabelling, like CD31 for example. In my opinion, authors aim to describe a tumor not to make its immunolabelling characterization. For immunolabeling characterization, authors would need to add many more immunohistochemical markers.
Comments on the Quality of English Language
Overall the manuscript would benefit from a review by a native English speaker. Some sentences are too long. Examples: lines 153-155 and 177-182 in discussion.
In lines 175-176 in discussion, the authors should rephrase the sentence. What do they want to say with the expression "were accepted"?
Author Response
Dear reviewer
Thank you for taking the time to review our manuscript.
Below you will find the point-by-point responses to your corrections / suggestions.
Lines 61-62: include information about packed cell volume
The authors followed the suggestion.
Lines 90-92: indicate why the pathologist had to perform immunohistochemistry. include full name before PAS acronym. Revise this aspect throughout the manuscript for other acronyms.
The authors followed the suggestion.
Immunohistochemistry: Explain the type of validation you made for c-kit and VEGF (if any). Include the positive and negative controls you used for all the markers used. See additional comments in the discussion for c-kit and VEGF.
The authors followed the suggestion.
Fig 2 (define PAS), ampliation, the value is the total or objective??? include arrows. Figure 3: include arrows and full name, not acronyms. What is IIC? 20x????
The authors followed the suggestion.
Results: what happened to the snake? Is he alive? Did he fully recover? The authors spend a lot of time on material and methods describing surgery and did not include any data about recovery and follow-up in the results.
The authors thanked the reviewer for this comment and decided to include the follow-up of the clinical case in the manuscript. “The animal was kept under observation for one day after surgery, after which it was returned to its owner. A telephone follow-up was carried out at 30 and 60 days post-surgery. In both follow-ups, the owner reported that the snake was in excellent health, feeding spontaneously and performing its usual activities.”
Discussion:
Lines 186-188 (about c-kit): This statement is a big limitation. I don't see the need to include the C-kit data. if so, you need to state the limitations and don't make any conclusion. What about if c-kit was positive? Would not be a carcinoma anymore? several carcinomas revealed c-kit positivity...
The authors followed the suggestion.
Lines 190-194: VEGF is not needed for diagnosis. In this context, VEGF immunolabelling doesn't bring any additional information, because you did not perform a new vessel immunolabelling, like CD31 for example. In my opinion, authors aim to describe a tumor not to make its immunolabelling characterization. For immunolabeling characterization, authors would need to add many more immunohistochemical markers.
The authors followed the suggestion.

Round 2
Reviewer 2 Report
Comments and Suggestions for Authors
The manuscript is much more clear now. However, there is still a need for improvement in figure legends, where acronyms should be avoided (AE1-AE3 in Figure 3, for example).
Author Response
Dear reviewer
Thank you for taking the time to review our manuscript.
Below you will find the point-by-point responses to your corrections / suggestions.
The manuscript is much more clear now. However, there is still a need for improvement in figure legends, where acronyms should be avoided (AE1-AE3 in Figure 3, for example).
The authors followed the suggestion.
